# Mechanistic basis of the dynamic response of TWIK1 ionic selectivity to pH

Franck C. Chatelain [1,2], Nicolas Gilbert [1,2], Delphine Bichet[1,2], Annaïse Jauch[3], Sylvain Feliciangeli[1,2], Florian Lesage [1,2] ✉ & Olivier Bignucolo [1,4]

Highly selective for K+ at neutral pH, the TWIK1 channel becomes permeable to Na+ upon acidification. Using molecular dynamics simulations, we identify a network of residues involved in this unique property. Between the open and closed states previously observed by electron microscopy, molecular dynamics simulations show that the channel undergoes conformational changes between pH 7.5–6 involving residues His122, Glu235, Lys246 and Phe109. A complex network of interactions surrounding the selectivity filter at high pH transforms into a simple set of stronger interactions at low pH. In particular, His122 protonated by acidification moves away from Lys246 and engages in a salt bridge with Glu235. In addition, stacking interactions between Phe109 and His122, which stabilize the selectivity filter in its K+-selective state at high pH, disappear upon acidification. This leads to dissociation of the Phe109 aromatic side chain from this network, resulting in the Na+-permeable conformation of the channel.

TWIK1 belongs to the family of K+ channels with two pore domains (K2P). TWIK1 has distinctive features: it is expressed in many tissues, it is constitutively endocytosed from the plasma membrane to recycling endosomes, and its ionic selectivity depends on variations in extracellular or intra-endosomal pH. In particular, a decrease in pH from 7.4 to 6 induces a decrease in K+ ion selectivity, allowing TWIK1 to conduct Na+ ions[1,2]. This change in ionic selectivity leads to a reduction in the net flow of positive charges, with the inflow of Na+ opposing the outflow of K+. This change in selectivity is slow and reversible[3,4]. In endosomes with intravesicular acidic pH, Na+-permeable TWIK1 could contribute to organelle acidification by helping to substitute Na+ for H+. At the cell surface, TWIK1 current could facilitate membrane hyperpolarization or depolarization, depending on its recycling rate between the plasma membrane and recycling endosomes[1,2]. Gene inactivation of TWIK1 in pancreatic and renal cells leads to an increase of the resting membrane potential, indicating that, in these cells, it behaves as a depolarizing channel permeable to Na+[1,2,5]. Hypokalemia has also been shown to promote a shift in TWIK1 ionic selectivity toward the Na+-permeable state. This dynamic selectivity underlies the

paradoxical depolarization of human cardiomyocytes during hypokalemia[1].

Active K2P channels are dimer of subunits, each subunit containing two pore domains (P1 and P2) that contribute to the formation of the ionic pore. TWIK1 has unique residues in these domains that are involved in its ionic selectivity. We have shown that the T118I mutation in the P1 domain suppresses extracellular pH sensitivity and dynamic selectivity, producing a channel locked in the K+-selective state[1]. We have also identified residues contributing to its sensitivity to pH: His122, which is the primary pH sensor, and Leu108 and Phe109, two pore helix residues. The H122N mutation abolishes TWIK1 sensitivity to acidification whereas the L108F/F109Y mutation produces a channel that is constitutively permeable to Na+[2]. Progressive acidification causes first a slight increase in TWIK1 current between pH7.5 to 6.5, then a decrease associated with a substantial change in ionic selectivity below pH6.5[2]. The current produced by the H122K mutant, which mimics the protonated state of the His122 side chain upon acidification, is already maximal between pH7.5–7.0, but still exhibits the ionic selectivity change below pH6.5[2] suggesting that His122 is not the only

[1]Université Côte d'Azur, Centre national de la recherche scientifique, Institut national de la santé et de la recherche médicale, Institut de pharmacologie moléculaire et cellulaire, 06560 Valbonne, France. [2]Laboratories of Excellence, Ion Channel Science and Therapeutics, 06560 Valbonne, France. [3]Immunodeficiency Laboratory, Department of Biomedicine, Basel, Switzerland. [4]Swiss Institute of Bioinformatics, Basel, Switzerland. ✉e-mail: lesage@ipmc.cnrs.fr

pH-sensitive structural element involved in this change. Previous studies have described the structures of TWIK1 in its open state at pH8[6] and 7.4[7], and in its closed state at pH5[7]. Using molecular dynamics (MD) simulations, a previous study highlighted the importance of Thr118 and Leu228, two residues not found in the selectivity filter of other $K_{2P}$ channels, for the dynamic selectivity of TWIK1[3]. However, how TWIK1 becomes permeable to $Na^+$ at pH6 has not yet been studied. To elucidate the molecular mechanism controlling the dynamic response of TWIK1 to pH, we use here a comprehensive approach combining mutagenesis and electrophysiology experiments with pKa calculations and molecular dynamics (MD) simulations.

## Results

### A method to identify residues involved in the response of TWIK1 ionic selectivity to pH

Progressive acidification of the extracellular pH has sequential effects on TWIK1. First, channel activity increases between pH7.5 and pH6.5, then ionic selectivity changes at lower pH levels as shown by the shift of the reversal potential (Fig. 1A–C)[2]. As observed with H122K[2], the H122R mutation that mimics the protonation of the His122 side chain leads to an increase in TWIK1 activity at high pH, without affecting the subsequent change in ionic selectivity at lower pH (Fig. 1D–F). This result shows that other residues are required for the acidification-induced change in ionic selectivity. To identify these residues we first focused on the area of the selectivity filter (Fig. 2). Mutations of TWIK1 residues were designed to test putative mechanisms within the limits of available natural amino acids (Fig. 3). For example, replacing histidine with an asparagine gives comparable side-chain volumes and hydrophilicity, while suppressing protonation at low pH. pKa calculation was performed on all titratable TWIK1 residues located in the crystal structure. These calculations enabled us to select the selectivity filter residues to be titrated for MD simulations (residues shown in blue in Fig. 2) and the trajectories of these simulations were analyzed to highlight alterations within the atomic networks (Fig. 4 and Fig. S1). The next sections describe how interactions between His122, Lys246, Glu235, Phe109 and Ser106 as well as Asp230 respond to acidification. These descriptions provide an atomic-level overview of pH-induced responses for most of these key elements.

### The Glu235-Lys246 interaction

Lys246 is located in the M4 membrane-spanning helix, while Glu235 is found in the extended, flexible loop connecting the SF2 and M4 domains (Fig. 2A, C and D). This SF2-M4 loop, stabilized by a network of hydrogen bonds, is known to be important for the function of TREK and TASK $K_{2P}$ channels[4,5]. Like H122R, the TWIK1.E235Q mutant shows maximal channel activity at pH7.5, with no activation phase between 7.5 and 7.0 (Fig. 3A–C). Notably, this mutation has no effect on the change of ionic selectivity below pH6, as shown by the shift in reversal potential between pH7.5 and pH6 ($\Delta E_{Rev} = 17.31 \pm 2.63$ mV) (Fig. 3A–C and Table 1). The ionic selectivity of the K246Q mutant is almost identical from pH7.5 to 6.0, resembling TWIK1 at pH7.5. However, below pH7, the current amplitude of the K246Q mutant increases and remains high at pH6, unlike TWIK1 (Fig. 3D–F). These results suggest that Glu235 is involved in the activation phase but not in the selectivity change, while Lys246 is involved in this change but not in channel activation. We then tested double mutations. The E235Q-K246Q mutation that disrupts the strong electrostatic interactions between these two residues eliminates both channel activation and ionic selectivity change (Fig. 3G–I). $E_{Rev}$ is stable over the entire tested pH range, showing that ionic selectivity is independent of pH (Table 1). To confirm the importance of electrostatic interactions in this region, the two residues were swapped in the TWIK1.E235K-K246E mutant. This channel shows a pH response comparable to that of TWIK1 with a change in ionic selectivity at low pH, with $\Delta E_{Rev}$ becoming significantly different below pH6 ($\Delta E_{rev} = 17.19 \pm 3.93$, Table 1) (Fig. 3J–L). These results demonstrate that Glu235 and Lys246 are involved in the pH sensitivity of TWIK1.

### The His122-Glu235-Lys246 triad

To understand how His122 interacts with Glu235 and Lys246, we performed MD simulations. Unbiased simulations were conducted at pH7.4 and pH6.0. Protonation states of the titratable residue were initialized on the basis of their calculated pKa values, generating ~12.0 μs of simulation distributed across 36 distinct protein trajectories, 18 at each pH value. Additional simulations with the TWIK1.H122N mutant were carried out. These simulations encompassed 16 independent protein trajectories, again 8 at each pH value,

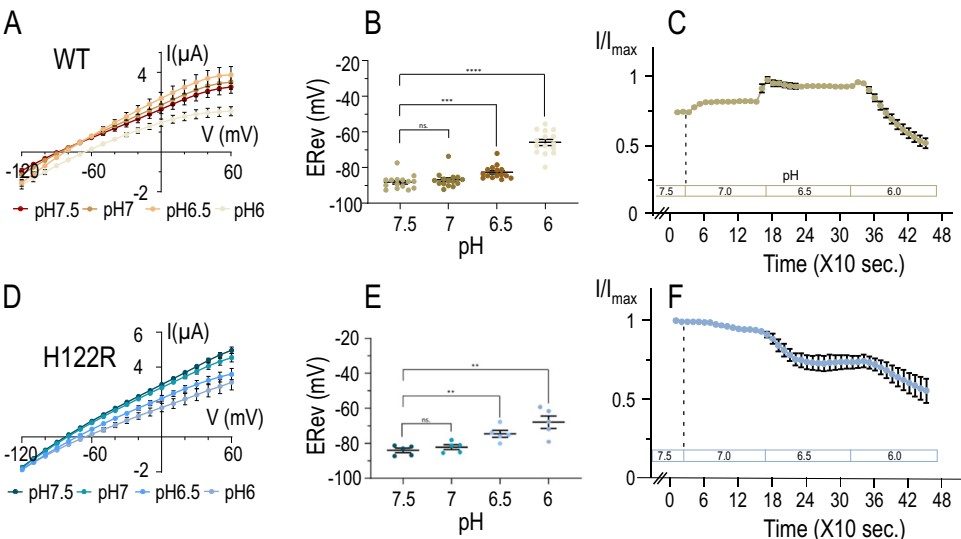

**Fig. 1 | pH-sensitivity of TWIK1.H122R ionic selectivity.** TWIK1 (in brown shades) and TWIK1.H122R (in blue shades) currents recorded in *Xenopus* oocytes at pH7.5, pH7, pH6.5 and pH6. After stabilization of the currents at pH7.5, voltage ramps were applied from −120 mV – +60 mV every ten seconds. pH was changed every 150 s starting from pH7.5–pH6. **A**, **D** I/V curves recorded during the stabilized currents at different pH. **B**, **E** Reversal potentials ($E_{Rev}$) extracted from the voltage ramp recorded at each pH. **C**, **F** Kinetics of the current variations as a function of the pH, measured at 0 mV every 10 s. pH changes are indicated. All data are presented as mean ± SEM; Statistical relevance has been evaluated using unpaired *t*-test: ns, non-significant ($p = 0.37$ (**B**) and $p = 0.35$ (**E**)); **$p < 0.01$ ($p = 0.0027$ and $0.0039$ (**E**)); ***$p < 0.001$ ($p = 0.0004$ (**B**)); ****$p < 0.0001$ (**B**). *n* is the number of oocytes examined for **A**–**C** ($n = 15$) and **D**–**F** ($n = 5$).

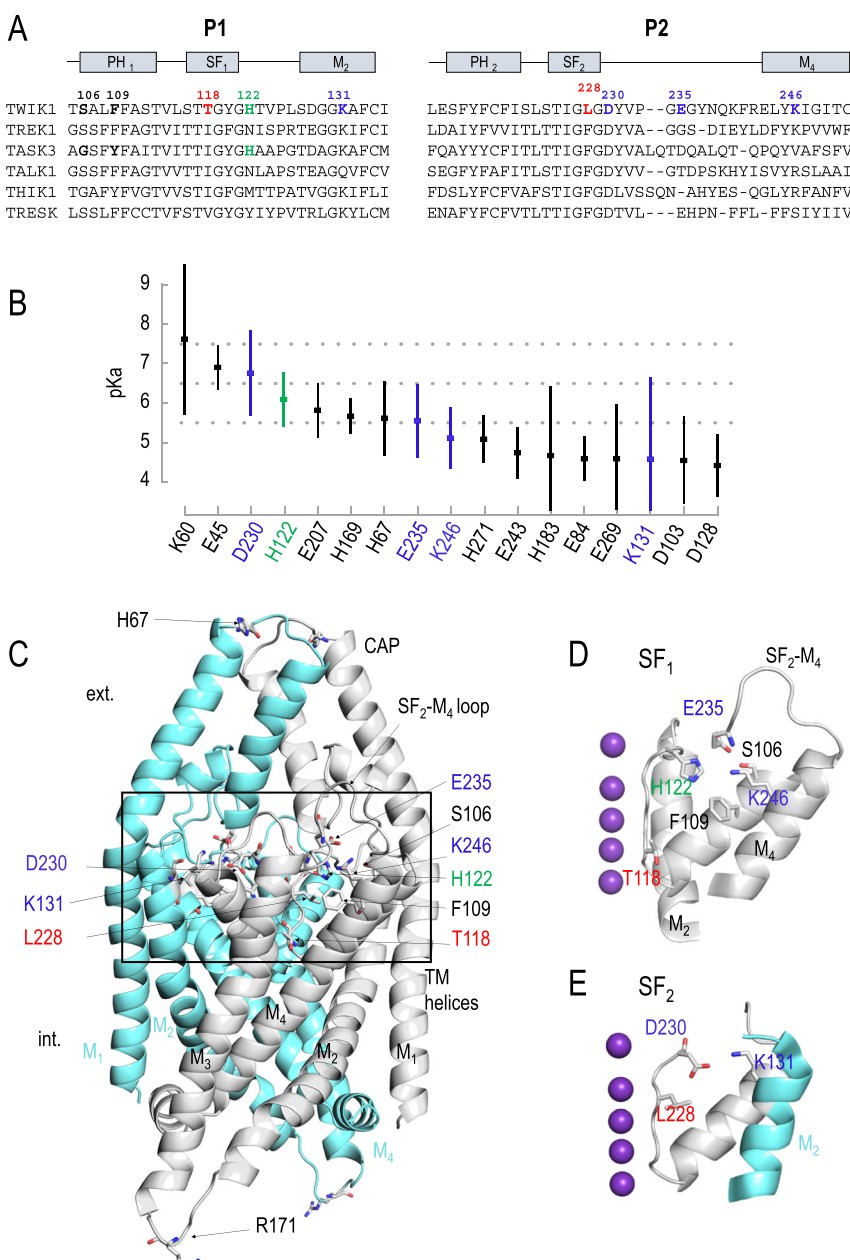

**Fig. 2 | Titratable TWIK1 residues and other residues highlighted in this study.** In red, the residues unique to TWIK1 and involved in ionic selectivity change. In green, the primary pH sensor in TWIK1 and TASK3. In blue, the titratable residues highlighted in this study. In bold black, residues swapped between TWIK1 and TASK3. **A** Sequence alignment of human K$_{2P}$ channels. PH, pore helix domain; SF, selectivity filter domain; M, membrane-spanning helix. **B** Calculated pKas of TWIK1 titratable residues are presented in decreasing order for the values between 8.0 and 4.0 (means ± standard deviations, $n = 12$ is the number of identical residues in the dimer (2), multiplied by the number of frames extracted from the MD (6)). **C** TWIK1 representation based on its crystal structure (PDB code 3UKM). TWIK1 subunits are shown in grey and cyan. Side chains of residues S106, F109, H122, L228, D230, E235, K246 of the same subunit converge towards the same region buried behind the selectivity filter domain 1 (SF1). The side chain of residue D230 of one subunit points in the direction of that of residue K131 of the neighbouring subunit behind the selectivity filter domain 2 (SF2). The residue R171 has been built using SWISS-MODEL into the PDB structure in order to appear in the structure. **D**, **E** Regions behind SF1 (**D**) and SF2 (**E**). In purple, K$^+$ ions in their respective S sites in the crystal structure.

for a total of 2.4 µs. The protonation pattern was identical, with the exception of residue 122. The region examined within the protein is illustrated in Fig. 4A, while Fig. 4B shows representative snapshots of typical interactions under the 4 conditions studied. Our data suggest that acidification gives rise to a salt bridge linking His122 and Glu235, and that the dynamics of this salt bridge may contribute to pH sensitivity (Fig. 4G). At pH7.4, this salt bridge is broken within a few tens of ns (Fig. S1). This pairwise analysis does not reveal a pH effect on the distance between His122 and Lys246 in TWIK1, whereas this distance is slightly increased by acidification in TWIK1.H122N (Fig. 4G). Due to the

low value of its calculated pKa (Fig. 2B), Lys246 was deprotonated and neutral at both pHs, whereas Glu235 remained unprotonated and negatively charged. However, the distance between these residues decreases significantly upon acidification (Fig. 4G and Fig. S1). As their protonation states are identical at both pH values, another interaction is required to explain this observation. The distance between Glu235 and Lys246 is not affected by pH in TWIK1.H122N, suggesting that protonation of His122 is necessary to induce this distance decrease. We have generated two-dimensional maps depicting distance distributions to analyze how distances between these three residues respond

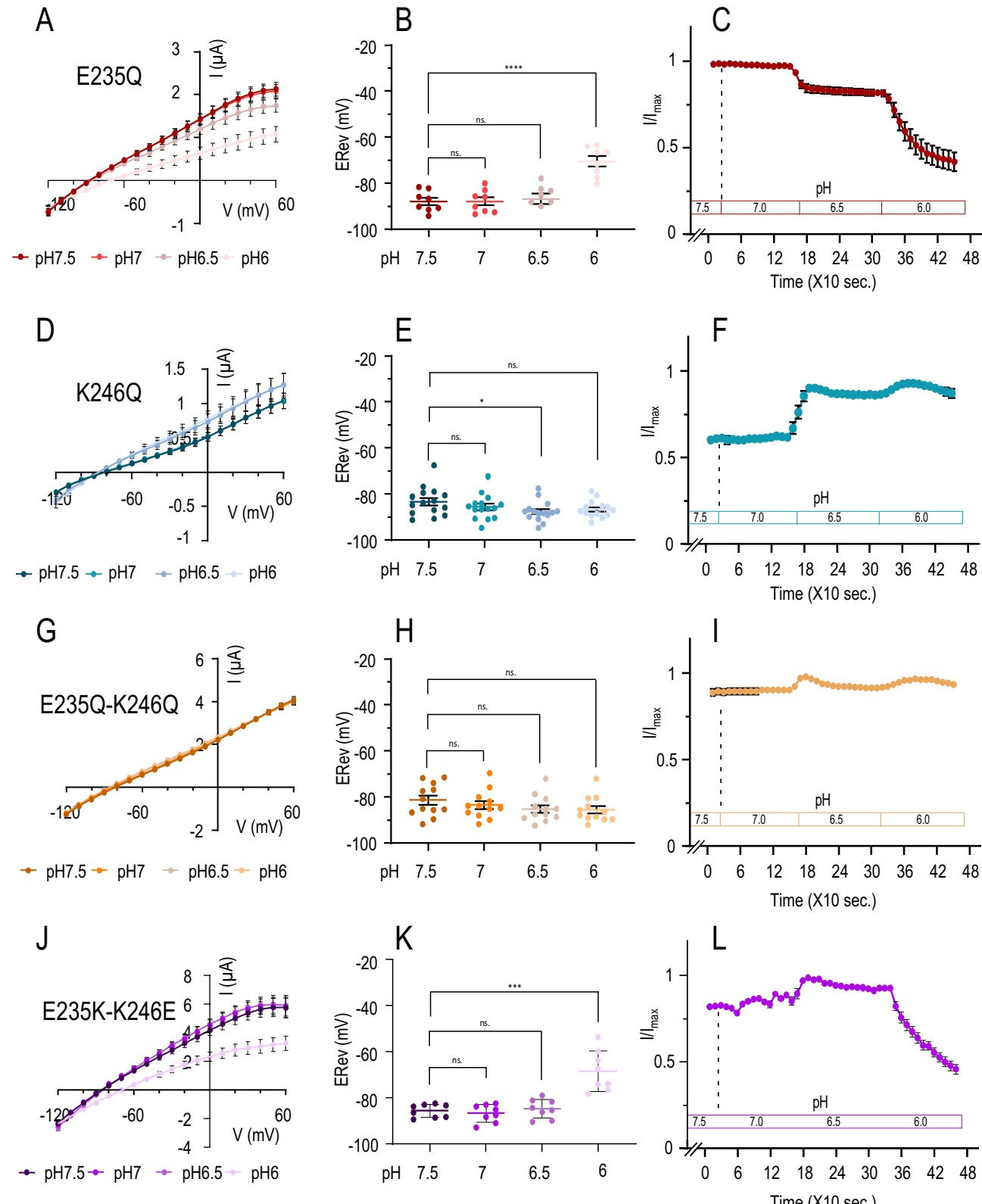

**Fig. 3 | pH response of currents produced by TWIK1-E235Q (in red shades), TWIK1.K246Q (in blue shades), TWIK1.E235Q-K246Q (in orange shades) and TWIK1.E235K-K246E (in violet shades).** After stabilization of the currents at pH7.5, voltage ramps were applied from −120 mV – +60 mV every ten seconds. pH was changed every 150 s. starting from pH7.5 to pH6. **A**, **D**, **G**, **J** I/V curves extracted from the stabilized currents. **B**, **E**, **H**, **K** Reversal potentials ($E_{Rev}$) measured during these stabilized currents and summarized in Table 1. **C**, **F**, **I**, **L** Kinetics of the current variations as a function of the pH, measured at 0 mV every 10 s. pH changes are indicated. All data are presented as mean ± SEM; Statistical relevance has been evaluated using unpaired *t*-test: ns, non-significant ($p = 0.68$ and $0.98$ (**B**), $p = 0.27$ (**E**), $p = 0.12$, $0.15$ and $0.43$ (**H**), $p = 0.54$ and $0.61$ (**K**)); *$p < 0.05$ ($p = 0.03$ (**E**)); ***$p < 0.001$ ($p = 0.0001$ (**K**)); ****$p < 0.0001$ (**B**). *n* is the number of oocytes examined for **A**–**C** ($n = 8$), **D**–**F** ($n = 15$), **G**, **H**, **I** ($n = 12$) and **J**–**L** ($n = 8$).

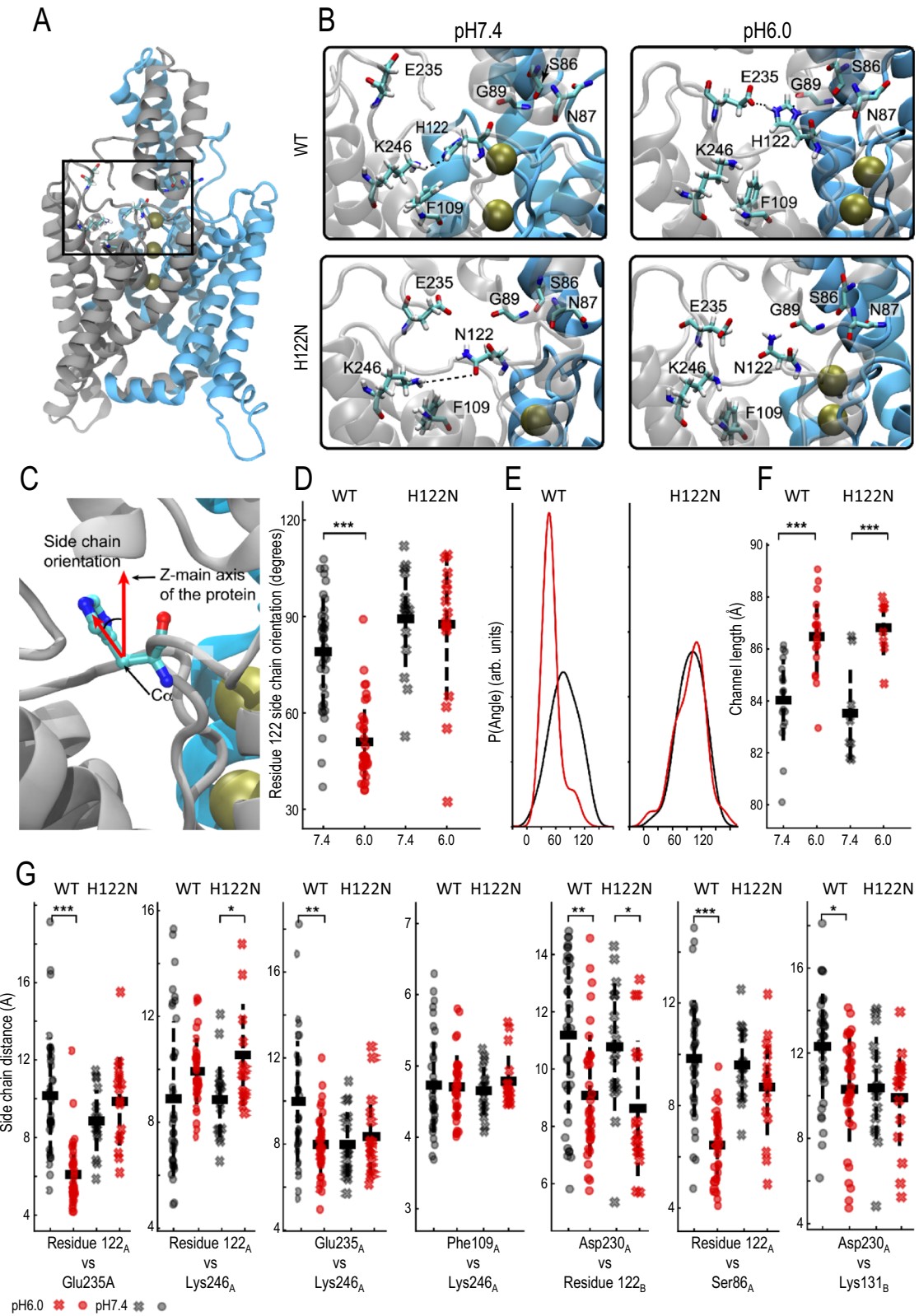

to pH and mutation. The frequency of occurrences is colour-coded using a gradient from violet to yellow, with shades from light green to yellow indicating regions of highest occurrence density (Fig. 5). We began by examining the Glu235-Lys246-His122 triad and analyzing the distance maps. Specifically, we plotted Glu235-Lys246 distances along the *X*-axis and residue122-Glu235 distances along the *Y*-axis (Fig. 5A). At high pH, three distinct regions emerge for TWIK1. The first graph from the left contains a greenish region labelled "a," indicating states where Glu235 interacts with Lys246 (~4 Å along the X-axis), while its interaction with His122 remains relatively weak. This greenish region corresponds to a distance between His122 and Glu235 ranging from 6.5–10 Å. A yellow region labelled "b" characterizes a short distance between His122 and Glu235 (around 5 Å) coupled with a substantial Glu235-Lys246 distance along the *X*-axis (8 Å). Notably, no data points

**Fig. 4 | MD simulations reveal low pH induced conformational changes.**
**A** Molecular representation of TWIK1. The subunits are coloured in blue and silver, and the studied region is indicated. The residues discussed in the text are shown as sticks. **B** Magnifications of the investigated region showing typical conformations adopted by the WT model at pH6.0 (upper left) and at pH7.4 (upper right) and by the H122N variant at pH6.0 (lower left) and at pH7.4 (lower right). Interactions are indicated by dotted lines. **C** Orientation of the residue 122 in respect to the channel principal axis (for details see the method section). **D** Orientation of residue 122 as a function of the pH and the identity of the residue. Each point represents the average of the values taken in each subunit over an independent 300-600 ns long simulation. Means, standard deviation and the results of an unpaired $t$-test assuming unequal variance and performed on 36 (WT) resp. 16 (H122N) pairs of values are indicated. $p = 2.493e^{-11}$ (WT) and 0.810 (mutant). **E** Histograms of residue 122 orientation fitted using the non-parametric Kernel density estimation function of the Python Pandas suite. All four distributions are unimodal. **F** Distance between the Cα atoms of His67 and Arg171, used as proxies for the length of the channel along its principal axis, are shown as a function of the pH and the identity of residue 122. For this analysis the appropriate replicate for the statistics is the whole channel,

i.e., the values of two subunits were pooled. Means, standard deviation and the results of an unpaired $t$-test assuming unequal variance and performed on 18 (WT) resp. 8 (H122N) pairs of values are indicated. $p = 4.182e^{-05}$ (WT) and 0.0009 (mutant). **G** Distances between the centres-of-mass of side chain heavy atoms of pairs of residues as a function of the pH and the identity of residue 122. Residue labels A and B differentiate between intra- and inter-subunit interactions. Each point represents the average of the values taken in each subunit over an independent 300–600 ns long simulation. Means (horizontal lines), standard deviations (vertical dashed lines) and the results of an unpaired $t$-test assuming unequal variance and performed on 36 (WT) resp. 16 (H122N) pairs of values are indicated. The results of statistical analyses are presented as follows: $*p < 0.05$; $**p < 0.01$, $***p < 0.001$. For simplification, $P$-values below 0.001 are not further identified, but are considered as $p < 0.001$. res122-E235: $p = 1.489e^{-09}$ (WT) and 0.173 (mutant); res122-K246: $p = 0.053$ (WT) and 0.0106 (mutant); E235-K246: $p = 0.00052$ (WT) and 0.547 (mutant); F109-K246: $p = 0.78$ (WT) and 0.30 (mutant); res122-D230: $p = 0.000589$ (WT) and 0.0157 (mutant); res122-S86: $p = 9.07e^{-11}$ (WT) and 0.158 (mutant); D230-K131b: $p = 0.00082$ (WT) and 0.301 (mutant).

**Table 1 | Key values from electrophysiological recordings of TWIK1 mutants**

| Construct | Current at $I_{0\ mV}$ (μA) | | | | ERev (mV) | | | | ΔERev (mV) between pH7.5 and pH6 |
|---|---|---|---|---|---|---|---|---|---|
| | pH7.5 | pH7.0 | pH6.5 | pH6 | pH7.5 | pH7.0 | pH6.5 | pH6 | |
| TWIK1 (n = 15) | 2.15 ± 0.23 | 2.36 ± 0.26 | 2.69 ± 0.29 | 1.47 ± 0.17 | −88.21 ± 0.99 | −86.87 ± 1.092 | −82.64 ± 0.98 | −65.83 ± 1.69 | 22.37 ± 1.56 |
| T118I (n = 6) | 3.20 ± 0.86 | 3.54 ± 1.06 | 2.87 ± 1.14 | 4.30 ± 1.26 | −79.67 ± 3.2 | −82.67 ± 3.13 | −83.67 ± 2.43 | −84.17 ± 2.13 | −4.5 ± 1.64 |
| L228F (n = 14) | 1.27 ± 0.46 | N/A | N/A | 1.20 ± 0.40 | −93.35 ± 2.17 | N/A | N/A | −93.93 ± 2.21 | −0.58 ± 1.15 |
| F109Y (n = 8) | 0.19 ± 0.02 | N/A | N/A | 0.13 ± 0.01 | −57.9 ± 4.25 | N/A | N/A | −50.22 ± 6.26 | 6.87 ± 2.84 |
| H122N (n = 6) | 1.09 ± 0.31 | 1.08 ± 0.35 | 1.034 ± 0.31 | 0.987 ± 0.30 | −67 ± 1.79 | −67.5 ± 0.95 | −66.25 ± 0.62 | −64.75 ± 0.75 | 2.25 ± 2.17 |
| D230N (n = 6) | 0.32 ± 0.10 | 0.36 ± 0.11 | 0.34 ± 0.11 | 0.33 ± 0.11 | −20.01 ± 3.21 | −18.3 ± 2.69 | −16.08 ± 2.17 | −14.79 ± 2.12 | 5.22 ± 1.55 |
| K131Q (n = 14) | 4.54 ± 0.45 | 5.21 ± 0.52 | 5.766 ± 0.56 | 3.36 ± 0.32 | −88.47 ± 0.94 | −88.45 ± 1.00 | −87.66 ± 0.98 | −81.47 ± 1.55 | 6.99 ± 1.13 |
| S106G (n = 6) | 1.42 ± 0.18 | 1.52 ± 0.19 | 1.59 ± 0.24 | 1.57 ± 0.29 | −87 ± 1.5 | −87.29 ± 1.62 | −82.88 ± 1.78 | −79.12 ± 2.41 | 7.88 ± 2.00 |
| H122R (n = 6) | 3.0 ± 0. 05 | 2.83 ± 0.07 | 2.21 ± 0.15 | 1.68 ± 0.24 | −84.00 ± 1.27 | −82.16 ± 1.37 | −74.51 ± 1.99 | −67.8 ± 3.54 | 16.14 ± 3.44 |
| E235Q (n = 8) | 1.43 ± 0.13 | 1.41 ± 0.13 | 1.19 ± 0.12 | 0.63 ± 0.14 | −87.75 ± 1.59 | −87.72 ± 1.74 | −86.63 ± 2.22 | −70.44 ± 2.27 | 17.31 ± 2.63 |
| K246Q (n = 17) | 0.51 ± 0.06 | 0.51 ± 0.06 | 0.73 ± 0.01 | 0.77 ± 0.01 | −83.21 ± 1.60 | −85.60 ± 1.38 | −87.54 ± 1.12 | −86.61 ± 0.93 | −3.39 ± 1.13 |
| E235Q-K246Q (n = 12) | 2.20 ± 0.26 | 2.22 ± 0.26 | 2.28 ± 0.28 | 2.33 ± 0.28 | −81.17 ± 2.01 | −83.27 ± 1.74 | −84.97 ± 1.62 | −85.32 ± 1.59 | −4.15 ± 1.81 |
| E235K (n = 10) | 1.96 ± 0.14 | 1.30 ± 0.14 | 0.79 ± 0.11 | 0.342 ± 0.03 | −84.58 ± 1.09 | −76.27 ± 1.32 | −59.58 ± 2.30 | −32.59 ± 2.38 | 51.98 ± 2.42 |
| K246E (n = 11) | 0.88 ± 0.10 | 1.03 ± 0.11 | 0.96 ± 0.10 | 0.53 ± 0.05 | −86.65 ± 0.87 | −72.73 ± 1.01 | 58.81 ± 1.13 | −44.14 ± 1.34 | 42.51 ± 1.66 |
| E235K-K246E (n = 8) | 4.18 ± 0.37 | 4.49 ± 0.41 | 4.59 ± 0.37 | 2.32 ± 0.33 | −85.58 ± 1.00 | −86.64 ± 1.35 | −84.68 ± 1.44 | −68.38 ± 3.11 | 17.19 ± 3.93 |

In brackets the units of measurement (μA for the currents and mV for the imposed potential) or the number of *Xenopus* oocytes recorded per construction. ΔERev reflects the shift of the reversal potentials between pH7.5 and pH6. N/A stands for non-available.

correspond to simultaneous Glu235-Lys246 and His122-Glu235 interactions. This observation suggests that at pH7.4, Glu235 engages either with Lys246 or with His122, but not both simultaneously. In the third major conformation visited at this pH, Glu235 interacts with neither His122 nor Lys246 ("c"). At acidic pH (Fig. 5A, second graph from the left), this complex pattern disappears: almost all conformations converge on a region typical of a salt bridge connecting His122 and Glu235. Intriguingly, a small subset of conformations appears in which all three residues simultaneously interact (« d »), a conformation that is totally absent at high pH. While the low pH-induced salt bridge between Glu235 and His122 is detectable in Fig. 4E and Fig. S1, its precise co-occurrence with a distance of ~8 Å between Glu235 and Lys246 (region « e » in Fig. 5A) could not be guessed. This remarkable collapse of a multifaceted histogram to a singular state upon acidification is not seen in the simulations in which Asn122 replaces His122 (Fig. 5B). In Fig. 5C, D we represented this triad of residues by plotting the Glu235-Lys246 distances on the X-axis and the His122-Lys246 distances on the *Y*-axis. These graphs reveal that two populations of fairly short identical His122-Lys246 distances overlap in TWIK1 at neutral pH (Fig. 5C).

As above, for TWIK1 and TWIK1.H122N, the distribution converges toward a simpler representation at low pH, characterized by an increase in the distance between His122 and Lys246 (Fig. 5C, D)

## The dissociation of Phe109 from His122 at low pH contributes to the loss of ionic selectivity

We have previously demonstrated the importance of residue Phe109 in the change of selectivity associated with pH[2]. In the crystal structure resolved at pH8[6] and the cryogenic electron microscopy (Cryo-EM) structure at pH7.4[7], the side chains of residues Phe109, His122, Gly235, Lys246 and Ser106 form a pocket (Fig. 2D and Fig. 4B). The distance between Phe109 and Lys246 is not affected by pH (Fig. 4G) whereas the distance between His122 and Phe109 is significantly increased at low pH, as shown in Fig. 5E, X-axes. The two-dimensional plots of the distances between Phe109-His122 and His122-Glu235 side chains suggest that at high pH, most of the conformations corresponding to a large His122-Glu235 distance (Y-axis) coincide with His122 lying ~6 Å from Phe109 (X-axis), likely forming aromatic stacking interactions (Fig. 5E). Once again, the complex distribution at neutral pH reduces to a much

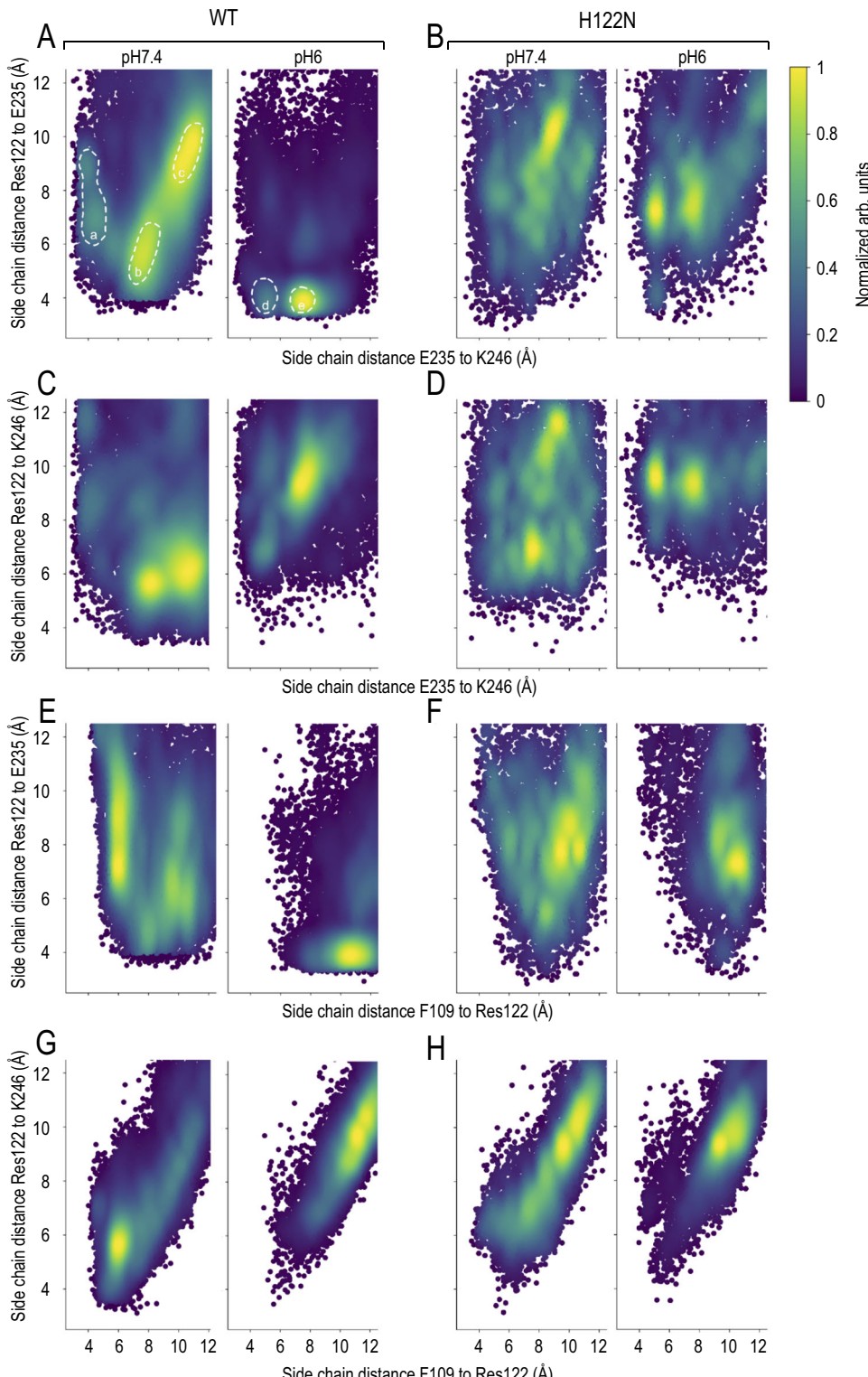

**Fig. 5 | Distance landscape representations reveal the dynamics of the His122-Glu235-Lys246 triple interaction and the Lys246-Phe109 movements.**
**A** Distances between the side chains of Glu235 and Lys246 (X-axis) and between His122 and Glu235 (Y-axis) at high and low pH. The normalized frequencies of occurrences are shown, yellow indicating the highest densities. The populated areas discussed in the text are indicated. The pH6.0 plot indicates that the His122- Glu235 salt bridge, identified in Fig. 5E, occurs essentially when the Glu235 and Lys246 are at ~8 Å from each other. **B** Similar distance maps for the H122N mutant. **C**, **D** Similar distance maps with the Y-axis corresponding to the distances between His122 (**C**) or Asn122 (**D**) and Lys246. **E**–**H** As above, whereas the X-Axis shows the distances between residue 122 side chain and the centre of mass of the Phe109 phenyl-ring.

simpler pattern at low pH, where most conformations correspond to the aforementioned His122-Glu235 salt bridge. This salt bridge is incongruent with concurrent aromatic stacking interactions, as many conformations converge into a solitary population with distances between aromatic rings spanning around 11 Å (Fig. 5E). The H122N variant does not show such a marked effect (Fig. 5F). These data suggest that, at high pH, His122, which is oriented at almost 90 degrees to the main axis of the protein, favours aromatic staking interactions with

Phe109, whereas the reorientation of its protonated form at low pH, coinciding with its attraction by Glu235, disrupts aromatic interactions with Phe109. The hydrophobic nature of Phe109 prevents it from following the upward movement of the protonated His122, while it can still make contact with Lys246, forming staking interactions with the hydrophobic methylene groups of the lysine side chain[8]. Consequently, the distance between Lys246 and His122 increases under low pH conditions, in line with our observations (Fig. 4E, Fig. 5C and Fig. 5G). From a mechanical point of view, the release of Lys246 and Phe109, rendering this large hydrophobic residue available for other functions at the level of the central cavity, would contribute to the decrease in ion selectivity. This mechanism relies on the hydrophobic nature of phenylalanine. A tyrosine at this position could have less impact, as it could maintain weak interactions with His122, even in its protonated state. This is supported by the lack of pH sensitivity of the F109Y mutant (Fig. S2A–D). TWIK1 and the related K$_{2P}$ channel TASK3 have a histidine at analogous positions (His122 and His98, respectively), whereas TASK3 has a tyrosine at position 85 instead of Phe109 in TWIK1 (Fig. 2A)[9,10]. Currents generated by the Y85F mutant of TASK3 were tested at pH7.5 and pH6. In contrast to TASK3, the TASK3.Y85F shows a shift in reversal potential similar to that of TWIK1, confirming the role of Phe109 hydrophobicity in ionic selectivity change upon acidification (Fig. S3). We evaluated another difference between TWIK1 and TASK3 in the degree of hydrophobicity close to the pH sensor. In the upper part of the first pore-helix domain of TWIK1, the side chain of Ser106 points towards His122 (Fig. 2D), whereas in TASK3, a glycine (Gly82) occupies the equivalent position (Fig. 2A). Whereas the S106G mutation in TWIK1 reduces the impact of acidification on ionic selectivity ($\Delta$ERev = 7.88 ± 2 mV vs. 22.37 ± 1.56 for TWIK1, Table 1, Fig. S2E), the G82S mutation in TASK3 is marked by an increase in current inhibition associated with a much more pronounced shift in reversal potential (Fig. S3), again confirming the importance of a hydrophilic residue at this position (106 in TWIK1) for the sensitivity of ionic selectivity to pH.

### At low pH, Asp230 stabilizes interaction between the two subunits in the TWIK1 dimer

Asp230 has a calculated pKa of around 6 (Fig. 2B). Its replacement by an asparagine in the D230N mutant produces a Na$^+$-permeable channel at pH7.5 (Fig. S4A). The proximity between the residue Asp230 of one subunit and the residue Lys131 of the adjacent subunit in the crystal structure resolved at pH8.0 suggests the formation of a salt bridge (Fig. 2E). This salt bridge would be disrupted by acidification and protonation of Asp230 favouring the Na$^+$-permeable state as observed with the TWIK.D230N mutant. However, replacing Lys131 with glutamine, a neutral residue of similar size and hydrophilicity does not suppress pH sensitivity, although the shift in reversal potential is less than in TWIK1 (TWIK1.K131Q mutant, Fig. S4B–D). These results underline the importance of these two residues, but rule out the existence of a salt bridge at high pH. Furthermore, MD simulations suggest that such a salt bridge between Asp230 of one subunit and Lys131 of another subunit forms essentially at low pH (Fig. 4G) in the WT channel. They also indicate that the distance between the carboxylic acid of Asp230 of one subunit and the imidazole of His122 of an adjacent subunit decreases significantly upon acidification and His122 protonation (Fig. 4G). The distance converges within 50 ns in both TWIK1 and TWIK1.H122N (Fig. S1). The same trend was observed for the distance between Asp230 of one subunit and Glu235 of the adjacent subunit.

In a previous MD study investigating the responses of the selectivity filter to various protonation states of His122 and Asp230, the authors observed great variability in the distance between the carboxylic acid of Asp230 and the phenolic oxygen of Tyr217, a residue located behind P2. However, their data could not identify a relationship between a protonation state and this distance[3]. According to our

simulations, this distance reduces at low pH (Fig. S5), a trend also observed in the case of the H122N mutant. As Asp230 is the only residue titrated in this region, we deduce that this pH response is related to the protonation of Asp230, probably in connection with the enlargement of the S0-S1 region.

These data suggest that, upon acidification, Asp230 stabilizes interactions between the two subunits of the TWIK1 dimer and approaches Tyr217, located behind the selectivity filter.

### Low pH promotes stronger interactions between His122 and the lower part of the CAP

The model of the membrane-inserted TWIK1 protein used for MD simulations was built from the 2012 structure identified by PDB accession code 3UKM[6]. We compared the structures generated by MD simulation with those obtained more recently by cryo-EM at pH7.4 and 5[7]. In the cryo-EM structures, the orientation of His122 with respect to the main axis of the protein decreases significantly upon extracellular acidification[7]. For each frame, the protein was oriented along its major axis and the angle formed by the major axis and a line joining the C$\alpha$ atom and the side chain of residue 122 was calculated. On average, the orientation of His122 relative to the major axis of the protein decreases from ~80 to ~50 degrees upon acidification. Interestingly, no response is observed with asparagine in the TWIK.H122N mutant, which remains at ~90 degrees at both pH values (Fig. 4C). Overall, probability histograms of the angle for Asn122 support unimodal distributions. In simpler terms, Asn122 rarely adopts a low-angle conformation, its distributions at both pH values being almost indistinguishable. Conversely, His122 exhibits a pronounced preference for a high-angle orientation at pH7.4 and a low-angle orientation at pH6. Notably, the simulations effectively replicated the behaviours inferred from the cryo-EM structures.

In terms of long-range conformational changes, the low pH structure is several Å longer than the high pH structure, mainly due to rearrangements of the helical CAP and transmembrane helices. To check whether long-range conformational changes are observable in the simulated structures, we selected the C$\alpha$ atoms of two residues spanning the entire protein structure: His67 in the apex of the CAP, and Arg171 in the intracellular loop connecting TM2 and TM3. We compared the distance between them under the four conditions. His67 and Arg171 are shown in Fig. 2C. This sampling was sufficient to observe statistically significant long-range responses, as illustrated in Fig. 4F, which shows that the channel exposed to low pH elongates significantly by 2 to 3 Å. This response is also observed with the TWIK1.H122N mutant, which was ~2 Å longer in the trajectories corresponding to low pH conditions. A further consequence of His122 repositioning during acidification is a displacement of residues in contact with His122. This is illustrated by the reduced distance between the side chains of His122 and Ser86 under low pH conditions (Fig. 4G). While the centers of mass of their functional group are separated by around 10 Å at neutral pH, they move closer together by 6 Å at low pH. This movement closely reflects the high (13 Å) and low pH (9 Å) distances between these groups recorded in cryo-EM structures[7]. A comparable decrease in distance upon acidification was also observed for Asn87 and Gly89. These distances were studied to highlight interactions between His122 and the lower segment of the CAP. Our results indicate that at low pH, His122 engages more frequently with residues of the Extracellular CAP helix 2 (EC2). This behaviour is absent in the H122N mutant, confirming the idea that changes in pH have no impact on Asn122 orientation (Fig. 4C–E and Fig. 4G).

## Discussion

This study reveals unique features in the TWIK1 pore that underly its response to pH. Acidification of the extracellular medium from pH7.5–pH7 causes protonation of His122 and an increase in K$^+$ current.

Then, for pH below 6.5, a second step involving residues located in the pore region leads to a change of ionic selectivity. We had already identified some of the residues involved in the dynamic selectivity of TWIK1, notably Thr118 in the SF1 signature sequence, and His122 and Phe109, located respectively at the extracellular exit of the selectivity filter and in the SF1 pore-helix. Here, we identify Glu235 and Lys246 whose interactions with the main proton sensor His122 are crucial for the ionic selectivity change. Both residues are situated in the SF2-M4 loop, at the interface between the P1 and M4 regions. Their side chains occupy the same space as His122, Phe109 and the polar residue Ser106, and interact with them. MD simulation suggests that during the protonation process, His122 ceases to interact with Phe109 and, in a movement of its side chain toward the CAP, establishes a stable salt bridge with Glu235 in the SF2-M4 loop. This pair transforms into a triad with Lys246 located in the M4 helix, which approaches Glu235, forming rare conformations in which all three residues interact. Glu235 and Lys246 have low calculated pKa values, so that the interaction mentioned here describe a case in which three residues share a proton. These data suggest that, at pH values slightly lower than the ones considered here (pH < 6), Glu235 and Lys246 would share an additional proton. In this case, the triple interaction could strengthen, restraining the two M4 helices closer to the pore, in a mechanism reminiscent to the previously described KvAP potassium channel deactivation[11]. The replacement of Glu235 by a non-titratable residue in the E235Q mutation or its interaction with protonated His122 seems to place the channel in a state of maximal activity at a pH value above pH6.5. This state of maximal activity persists at pH values >6.0, for K246E and E235Q-K346Q mutants, all three conditions preventing the formation of the interaction Lys246-Glu235-His122 mentioned above. The importance of this channel region has already been highlighted in the description of K⁺-dependent gating of the TREK1 $K_{2P}$ channel[5]. In a study combining data from crystallography, electrophysiology and MD simulations, the authors showed that under low K⁺ conditions, the selectivity filter undergoes a significant reorganization of its architecture at the S1 and S2 sites as well as a large movement of the SF2-M4 loop that results in the inhibition of ion conduction. The binding of a specific TREK1 activator in the P1-M4 interface of this channel limits the movement of the SF2-M4 loop and stabilizes the channel in an open conformation[5]. A recent study of the pH–dependent gating of the alkaline-activated TALK1 channel also highlighted this region as essential for channel inactivation[12]. TALK1 contains an arginine residue in place of Lys246 in TWIK1, and its activation occurs at a remarkable high pH[13]. The authors found that Arg233 of TALK1 acts as a main proton sensor, its deprotonation triggering a sequence of allosteric changes that stabilize the S1 site of the filter in a conductive state. Notably, activation curves of TALK1 and R233K mutant show an acidic shift of about 3 pH units in the latter case, which mimics Lys246 of TWIK1. Compared with the TREK1 and TALK1 channels, TWIK1 is unique in that acidification induces a conformational rearrangement of its pore region that first alters its ion selectivity before leading to channel closure.

Previously reported TWIK1 structures have all been obtained under high K⁺ concentrations, showing an open state at pH7.4 or 8 and a close state at pH5. Several studies have shown that K⁺ ion concentration significantly affects the gating, permeability and selectivity of K⁺ channels[5,14–16]. Our results show that when K⁺ concentration is close to the physiological level (2 mM), the regulation of TWIK1 by extracellular pH is much more complex than a simple switch from a conducting to a non-conducting state. Among the information provided by these structures, the most relevant is certainly the significant displacement of the two residues His122 and Asp230. The disorganization of pore K⁺-binding S sites at acidic pH, which probably results from the reorientation of these two residues but also from the weakening of the interaction forces located behind the selectivity filter, is reminiscent of the pore of the NaK channel whose conversion to a K⁺-selective channel depends on the presence of the four binding sites S1 to S4 capable of coordinating K⁺ ions[17]. Previous MD simulations of open channels suggest that protonation of His122 and Asp230 results in S sites of the selectivity filter filled with water molecules. This state of the pore would allow Na⁺ transport[3]. It is now generally admitted that, with the exception of TASK1 and TASK2 channels for which an additional inner gate has been described[18,19], the selectivity filter of the $K_{2P}$ channels is the main site of regulation of their activity, a regulation known as C-type gating in the other families of K⁺ channels[4]. TWIK1 also exhibits C-type gating. However, TWIK1 has outstanding features at the pore level. With two non-conserved residues in the P domains and multiple residues sensitive to pH in crucial positions, TWIK1 responds in a unique way to proton modulation. In most $K_{2P}$ channels the action of pH is monophasic, a decrease of the activity for TREK1, TASK1/2/3 and TALK1/2 channels and an activation for TREK2[13]. TWIK1 has a biphasic response: its activity augments when exposed to a modest pH drop, then its ionic selectivity changes for stronger acidifications. Our results do not allow us to conclude on the level of conduction of TWIK1 at pH lower than pH6, the pore could be in a non-conductive state as proposed by the cryo-EM at pH5.5 and 150 mM K⁺.

TWIK1 is conductive and non-selective at pH6 and predominantly distributed in recycling endosomes suggesting that TWIK1 could play a dual role as a classical K⁺-selective channel at the plasma membrane, and as a K⁺ and Na⁺-permeable channel in endosomes. Its recycling rate between the surface and the endosomes may therefore modulate its selectivity and its influence on the resting membrane potential. A pharmacological compound acting on the selectivity filter of TWIK1 to stabilize the channel in either the K⁺-selective or Na⁺-permeable form would be a useful tool to decipher its physiological roles. Another interesting result is that this unique property of TWIK1 can be transferred to TASK3 by introducing only a few residues typical of TWIK1. In TASK3, Y85F and G82S mutant show a response to pH similar to that of TWIK1. TASK3 is known to be involved in cell proliferation and carcinogenesis[20,21]. Interestingly, a TASK3.G82S variant has been described in the COSMIC database, as a somatic mutation found in adenocarcinoma (COSM3897140, COSV57284824, ICGC (STAD-US): Gastric Adenocarcinoma - TCGA, US). This mutation was found in gastric tissues in which the channel exposed to acidic pH should be permeable to K⁺ and Na⁺. It would be certainly interesting to study how this selectivity change of TASK3 affects cancer development.

In conclusion, the identification of titratable residues, electrophysiology and MD simulations have identified dynamic changes controlling the conformation of the TWIK1 pore that could not be identified by electron microscopy alone. These changes involve a network of residues in the immediate vicinity of the selectivity filter, showing that Na⁺ transport may result from an evolution of the classical K⁺-selective pore. In addition, such an approach based on MD simulation could be of interest for studying regulations of other ion channels based on conformational changes acting on their level of activity.

## Methods

### Mutagenesis and oocyte expression

TWIK1 and its mutants were cloned into the HindIII–XhoI sites of a derivative of pGEM vector. All channels used in this study were designed from the active form of TWIK1 in which the internalization and inhibitory intracellular basic motifs were mutated (mutations K274E, I293A, and I294A)[2]. Site-directed mutagenesis was performed based on PCR of the full-length plasmid by using Pfu Turbo DNA polymerase (Stratagene). The entire cDNA was sequenced. TWIK1 and TWIK1 mutant cDNAs were linearized at the AflII restriction site, and capped cRNA were synthesized by using the AmpliCap-Max™ T7 High Yield Message Maker Kit (CellScript™). Stage V–VI Xenopus laevis oocytes were collected, injected with 5–10 ng of each cRNA, and maintained at 18 °C in ND96 solution (96 mM NaCl, 2 mM KCl, 1 mM MgCl2, 5 mM HEPES, pH7.4).

## Electrophysiology

Oocytes were used 1–3 days after injection. Macroscopic currents were recorded with two-electrode voltage clamp (TEV 200 amplifier; Dagan). Electrodes were filled with 3 M KCl and had a resistance of 0.5–2 MΩ. A small chamber with a perfusion system was used to change extracellular solutions and was connected to a ground by a 3 M KCl agarose bridge. All constructs were recorded in ND96 at designated pH adjusted with NaOH. For pH-sensitivity experiments, all recordings were done in standard ND96 solution, adjusting pH with NaOH to values of 6 – 7.5. For every pH, the solution was buffered with 5 mM MES and 5 mM HEPES, and the pH was adjusted with NaOH 2 N. Currents were evoked from a −80 mV holding potential followed by a 300 ms ramp from −120 – +60 mV. All recordings were performed at 20 °C. Stimulation, data acquisition, and analysis were performed using pClamp software (Molecular Devices). The graphs, statistical analyses and figures related to the electrophysiology have been realized on the GraphPad Prism software. Statistical significance was determined by using unpaired Student's $t$-test and indicated as follows: $*P < 0.05$; $**P < 0.01$; $***P < 0.001$. $****P < 0.0001$

## Computational section

The inter-residue distance calculations reported in Table 1 and the 3D-structure representations S1 and S2 have been realised on PyMOL software[22] from the crystal structure of TWIK1 (PDB ID: 3UKM, chains A and B, https://doi.org/10.2210/pdb3UKM/pdb). The 3D-structure representations reported in Fig. 4 were realised with VMD (version 1.9.4a51)[23]. Other structural features reported were calculated using a combination of GROMACS[24] tools and Python scripts. Where appropriate, multifactorial ANOVA were conducted. Landscape representations are essentially smoothed surfaces of distance occurrences. To construct them, the distances to be studied were matched: a given His122-Glu235 distance occurs at the same time as a given Glu235-Lys246 distance, enabling us to construct grids and 2D histograms of the occurrences. These data were then smoothed using the Scipy's splinef2d method. For this, Python 3.8.5 was used, in which the subpackage scipy.interpolate from the scipy version 1.5.2 was imported. The Python script that generates the landscape representations and the distance files are provided with this article.

## pKa calculations

pKa calculations are often performed uniquely on the available structure, so that conformational changes occurring upon proton binding or unbinding cannot be estimated. Also, changes in protonation states of a given residue affect its electrostatic environment, which may further modify the probability of a neighbouring residue to bind or release a proton. These issues are recognized, and it is tried to solve them implicitly through rearrangement of charges as a function of the dielectric constants of the protein and the solvent. Yet, these strategies were shown to fail when facing complicated proteins[25,26]. In this work, we aimed at overcoming these limitations by designing a strategy that, while using the state-of-the-art CHARMM's Poisson-Boltzmann solver to guide us in the choice of protonation states, involves several successive short MD simulations harbouring progressive protonation states until convergence. The atomic model of the TWIK1 channel was initially built on the basis of the 3UKM crystal structure[6], which is assumed to correspond to an open state, and completed for short missing loops using SWISS-MODEL[27]. The model was inserted in a 1-palmitoyl-2-oleoyl-sn-glycerol-3-phosphocholine (POPC) bilayer (≈230 molecules) and solvated (≈55000 TIP3P water molecules[28]) at 0.15 M KCl, resulting in a 91 × 91 × 130 Å box. A first 10 ns long simulation was conducted, and frames extracted at 2 ns interval for pKa calculations using a solver of the Poisson-Boltzmann (PB) equation. The length of the simulation was chosen to be sufficiently long to induce some relaxation of the biomolecules while staying short enough to avoid large conformational changes of the protein. The CHARMM module PBEQ (Poisson-Boltzmann equilibrium) was used because of its accuracy and versatility. A comparison with the fast pKa automated system H++ supports the idea that the PBEQ module, although time consuming, is sufficiently accurate for this type of project, as shown in supplementary section SI-1[29]. For each frame, the pKa were then calculated as done previously[30]. Briefly, after the first MD simulation and pKa estimation, the protonation state of titratable residues was updated as a function of the desired pH. A second MD simulation was then conducted, starting from the same initial configuration, but with these residues protonated. After two successive rounds, the MD simulation + pKa calculation were conducted until no more residues had to be modified. All glutamate, aspartate, lysine and histidine of TWIK1 residues were submitted to this process. Using this approach, we computed the pKa values for titratable residues, focusing on residues exhibiting values between 8 and 4 (Fig. 2B), many of which form a ring encircling the pore (Fig. 2C). A decreasing trend of pKa values emerged from the pore's core to its exterior. Subsequently, unbiased MD simulations were conducted to replicate systems at pH7.4 and pH6.0. Titratable residue protonation states were initialized based on their pKa values, generating ~12.0 µs of simulation distributed across 36 distinct protein trajectories. Notably, due to the pH insensitivity of the H122N mutant[2,7] (Fig. S2A, C), additional simulations of this variant were executed. These simulations encompassed 16 independent protein trajectories, accumulating a total of 2.4 µs. The identical protonation pattern was maintained, with the exception of residue 122. Here we report the protonation states of the residues located close to the studied area, comprising all titratable residues located within 12 Å from the Cα atom of residue 122. In this surrounding, His122, Glu45, Glu49 and Asp230 were protonated at low pH only, Glu235 was left negatively charged in all cases, whereas Lys246 and Lys131 were deprotonated in all cases.

## Molecular dynamics simulations

We constructed models of the channel based on the crystal structure with code 3UKM, which was prepared at a pH >7.4 and is believed to represent an open conformation. These models were protonated to simulate either pH6.0 or pH7.4 and were embedded in a double bilayer system, following a previously established method[11]. It's important to note that this system was different from the one used for the pKas calculations. In each extracellular-facing leaflet of the bilayer, there were 105 molecules of 1-palmitoyl-2-oleoyl-sn-glycerol-3-phosphocholine (POPC) and 10 of 1-palmitoyl-2-oleoylphosphatidylethanolamine (POPE). To partially recreate the negative charge inside biological membranes, the cytoplasmic-facing leaflet contained 10 negatively charged dimyristoylphosphatidylglycerol (DMPG) molecules and 105 POPC molecules only. We show in SI-2 section that this membrane modification is unlikely to significantly affect the pKas of the membrane inserted protein. The numbers of lipids were chosen to ensure that both leaflets had a comparable surface area, considering the structure of the channel. After solubilisation in 150 mmol·L-1 KCl and neutralisation, a typical simulation box contained ∼ 243'000 atoms. For the wild-type (WT) sequence, we performed a total of 18 trajectories at each pH, resulting in a cumulative simulation time of 5.8 µs at pH 7.4 and 4.8 µs at pH6.0. A similar procedure was followed for the H122N mutant, except that in this case, we conducted four simulations, each lasting 300 ns, at each pH value. Therefore, the total trajectory time dedicated to the H122N mutant was 2.4 µs. The all-atom MD simulations were performed with the CHARMM36 force field[31] using the GROMACS package version 2021.5[24]. The TIP3P water model was used[28]. Bond and angle lengths involving hydrogen atoms were constrained using the LINCS algorithm allowing an integration time step of 2 fs. Short-range electrostatics were cut off at 1.2 nm. Van der Waals interactions were calculated explicitly up to 10 Å, beyond which a

switch function was used to smoothly switch off the interactions to reach zero at 12 Å. Long-range electrostatic interactions were calculated by the PME algorithm[32]. The protein, lipids, and water/ions were coupled separately to a temperature bath at 310 K with the Nose-Hoover method with a time constant of 1.0 ps[33,34]. The system pressure was kept constant by semi-isotropic Parrinello-Rahman coupling to a reference value of 1 bar as implemented in the GROMACS suite. GROMACS and in-house Python scripts were used to analyse the data. Time series analyses reveal that the Cα RMSD approached values close to 3 Å over the course of the simulations, but without achieving prefect convergence. More precisely, the interactions studied in this article converged significantly and rapidly, enabling us to incorporate all values after 50 ns of stabilisation in our calculations. Some distances involving contact losses required more time to converge (Fig. S1). To determine the orientation of residue 122 with respect to the main axis of the channel, the structures were aligned along their main axis. The vertical red arrow in Fig. 4C shows this orientation. A second vector links the Cα atom of residue 122 with the center of mass of the selected side chain pairs, which are ND1 and NE2 in the case of histidine and OD1 and ND2 in the case of asparagine. The angle between these two vectors determines the orientation of the residue.

## Reporting summary

Further information on research design is available in the Nature Portfolio Reporting Summary linked to this article.

## Data availability

The data that support this study are available from the corresponding authors upon request. A source_data.zip file is provided with this paper, which contains the data necessary to reconstruct the MD RMSD values, the distance and angles values reported in Fig. 4, the distances incorporated in Fig. 5A and C, and a Python script that generates the landscape representations. The source data for electrophysiology underlying Figs. 1, 3, and Supplementary Figs. S2, S3 and S4 are also provided. Source data are provided with this paper.

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

## Acknowledgements

This work was funded by the Agence Nationale de la Recherche (grants ANR-11-LABX-0015-01 and ANR ANR-22-CE14-0048-01 to FL), by the Fondation pour la Recherche Médicale (équipe labellisée FRM 2020 to FL), and the Swiss National Supercomputing Center (CSCS) under the project IDs s968, s1037, s1099 and s1223 to OB. We thank Solène Gibaud for excellent technical help, and Simon Bernèche and Daniel Minor for useful comments on the manuscript.

## Author contributions

F.C.C., N.G., D.B., A.J. and O.B. performed the experimental and computational work. S.F., F.L. and all the other authors contributed to experience design, data analysis and manuscript writing.

## Competing interests

The authors declare no competing interests.

## Ethics

The animals used for oocyte collection were treated and sacrificed in accordance with procedures filed with and accepted by the French Ministry of Agriculture under authorization #14012-20180302144339 v3.
