## [Peer Review File · Nature Communications]

Mechanistic basis of the dynamic response of TWIK1 ionic selectivity to pHReviewer #1 (Remarks to the Author):

The manuscript by Chatelain et al. used a combination of MD simulations and experimental assays to study the effect of the pH on the TWIK1 ionic selectivity. While this paper represents a step forward in understanding the molecular basis of the pH-dependent change of selectivity in TWIK1 channels, I have several concerns that should be addressed before publication:

1- First, it is not clear if the authors used MD simulations to narrow down the number of residues to mutate experimentally or if they used MD simulations to interpret the experimental results. If they are in the first case, what's the reason why the authors experimentally mutated only few residues whose pKa calculations are reported in Figure 2? For example, why didn't the authors mutate K131? If we consider the error bars, there are more amino acids that have a pKa equal to 6, thus being interesting to analyze. Otherwise, a thorough interpretation of the experimental results would be done by repeating the computational analysis also on computational systems which carry on the same experimental mutations.

2- In Figure 1 the authors state "Residues affecting ..." but the reason of this statement is unclear. In Panel A and B there are electrophysiological assays that confirm the dependence of TWIK1 function to the pH but no data on specific residues that could affect this behavior are shown. Moreover, what is the connection of panels A and B with panel C?

3- As the authors claim to introduce a new method for predicting the pKa (lines 358-359), the advantages of their approach should be described in a better way, especially from a quantitative point of view. I expect a comparison of their results with state-of-art tools/methods such as <https://doi.org/10.1093/nar/gki464>, <https://doi.org/10.1002/prot.22102> and <https://doi.org/10.1021/acs.jctc.7b00875>.

4- Why is the simulated system different from that used to predict the pKa (lines 390-391)? If one repeated the same pKa predictions in this system, would their values conserved or not? How will it affect the MD simulations results?

5- The authors should comment on their results in the view of <https://doi.org/10.1016/j.bpj.2016.07.009>.

6- Conclusions are missing, which would help the reader putting together the results which might seem otherwise scattered

In summary, while this work presents several interesting aspects, major changes are required before it could be published to better highlight the unifying aspects of the multifaceted analysis and tools deployed.

Minor comments:

1- The description of the simulations is quite confusing. The authors should add a table in the SI resuming all the simulations of both the sections "pKa calculations" and "MD simulations" to clarify the number of replicas and the corresponding length.

2- The authors should provide the RMSD profiles of the protein in each simulation to assess the convergence of their structures.

3- Figures should be ordered as they appear in the main text, e.g. Figure 4 appears before Figure 3. Then, their quality must be improved. For example, the labels in Figure 4 are too small to be read, there is a wrong "Figure 4" on the left/bottom and its title is too generic. The same for Figs 5 and 6 where the values on the density bar should be added. Each panel must be labeled with a letter, not the entire row as the authors did. The expressions "middle panel", "middle" should be avoided.

4- The computational procedure used to quantify the orientation of the residues 122 should be described in the Methods section, not in the caption of Figure 4.

5- I would add one or two sentences at the end of the "Discussion" that resume the main results of the present work and highlight its impact in future research.

6- In Figure 4, I would move the analysis of the side chain's rotation to the SI because this information is implicitly included in the distance landscape representations. Moreover, I would try to

represent all the panels of Figures 5 and 6 in a single figure with a structural representation on VMD that clarify the interactions reported in each panel. Lines 251-252: the ref is missing
Line 262: "strengthen" instead of "strengthens"
Lines 268-269: the ref is missing
Line 347: missing ref for PyMol software
Line 349: the version of VMD should be specified
Line 362: clarify the dimensions of the simulation box
Line 363: the model of the water molecules is missing
Line 402: there are missing refs
Line 408: the temperature is not specified

Reviewer #2 (Remarks to the Author):

Chatlain et al present a mechanistic investigation into the ion selectivity of TWIK1 in response to pH. Using a combination of currently available TWIK1 structures, MD simulations, mutagenesis and electrophysiology they elucidate with surgical precision a network of residues around the channel's selectivity filter that are responsible for this pH-dependent ion selectivity.

This referee has no major concerns with the experimental data and only minor concerns with the presentation at some points:

- It is not clear to me what the color-coding of residues in figure 2 means. I cannot find an explanation of why some residues are labeled in purple, red, black, or green and if this means anything.
- The network of residues in figure 4 B are hard to see. Coloring the helices lighter/ making them more transparent/ increasing the size of the specific residues shown would help.

We thank the reviewers for their positive appreciation of the work and their helpful comments. The manuscript was modified according to their comments and suggestions, which improved the quality of data presentation and the overall relevance of the article.

Reviewer 1

We thank reviewer 1 for his positive assessment of the general interest of the work. Our responses to her/his comments and suggestions are detailed below.

Major concerns:

1- First, it is not clear if the authors used MD simulations to narrow down the number of residues to mutate experimentally or if they used MD simulations to interpret the experimental results. If they are in the first case, what's the reason why the authors experimentally mutated only few residues whose pKa calculations are reported in Figure 2? For example, why didn't the authors mutate K131? If we consider the error bars, there are more amino acids that have a pKa equal to 6, thus being interesting to analyze. Otherwise, a thorough interpretation of the experimental results would be done by repeating the computational analysis also on computational systems which carry on the same experimental mutations.

Some residues were already known to be important for the ionic selectivity of TWIK1 (H122, T118, L228, F109), and they are all located in the channel region that controls the potassium selectivity of all potassium channels. We calculated the pKas of all the titratable residues in the TWIK1 crystal structure, but then focused our attention on the titratable residues located close to this region (residues colored blue in Fig. 2). This is now better explained in the text (lines 86 to 89, "*pKa calculation was performed on all titratable TWIK1 residues located in the crystal structure. These calculations enabled us to select the selectivity filter residues to be titrated for MD simulations (residues shown in blue in Fig. 2) and the trajectories of these simulations were analyzed to highlight alterations within the atomic networks (Fig. 4 and Fig. S1).*") and illustrated in the revised Fig. 2 in which we have transferred the K2P channel sequence alignment (Fig. 2A). We carried out simultaneous MD simulations and electrophysiological characterization of the mutated channels and found convergences leading to the proposed mechanism. Because they are not directly involved in the network of titratable residues involving H122, E235 and K246 located in the 1 SF1 selectivity filter domain, K131 and D230 located in the SF2 domain were studied last. Electrophysiological data suggest the absence of a salt bridge sensitive to a slight pH shift between these two residues (from 7.4 to 6, Figure S4), in agreement with the very low pKa calculated for K131 (between 4 and 5). In a previous MD work carried out by Domene's laboratory, the authors calculated the pKas of TWIK1 residues and thus modified their protonation states to mimic low and high pH, in the same way as in our work. They also titrated only H122 and D230, confirming our view that K131 is probably not titratable in the pH range studied (Oakes et al, 2016). Mutagenesis shows that these two residues, one in subunit 1 and the other in subunit 2, are both important for channel stability but do not appear to be involved in the change in ionic selectivity between pH 7 and 6. For this reason, we did not perform MD simulations.

2- In Figure 1 the authors state “Residues affecting ...” but the reason of this statement is unclear. In Panel A and B there are electrophysiological assays that confirm the dependence of TWIK1 function to the pH but no data on specific residues that could affect this behavior are shown. Moreover, what is the connection of panels A and B with panel C? Panel C of the original Figure 1, showing sequence alignment, has been moved to the revised Figure 2 where it is more informative. The title of the revised figure 1 has been changed to be more specific: "pH-sensitivity of TWIK1.H122R ionic selectivity". This figure shows that mimicking the protonation of H122 does not affect dynamic ionic selectivity, implying that other TWIK1 residues are required for the acidification-induced change in ionic selectivity. The central question of the study is which residues are involved.

3- As the authors claim to introduce a new method for predicting the pKa (lines 358-359), the advantages of their approach should be described in a better way, especially from a quantitative point of view. I expect a comparison of their results with state-of-art tools/methods such as <https://doi.org/10.1093/nar/gki464>, <https://doi.org/10.1002/prot.22102> and <https://doi.org/10.1021/acs.jctc.7b00875>.

We thank the reviewer for addressing this point. We do not claim to have developed a new theoretical method for pKa calculations. We use the CHARMM Poisson-Boltzmann equation solver for the calculations. The originality of our approach remains at the practical level: whereas PB solvers are applied by many investigators to a static structure at the start of a classical simulation (we discuss cpHMD below), we implemented short MD simulations and applied the PB solver to extracted frames. The aim here was to increase conformational sampling and, consequently, the reliability of extracted pKas on average. However, the actual estimation was carried out using the PBEQ solver from CHARMM. To reduce the risk of misunderstanding, we have reworded the lines introducing our strategy and moved the lines describing the protein construct further down to make this paragraph more coherent.

Here we discuss the three proposed references:

- [Citation 2 \(https://doi.org/10.1002/prot.22102\)](https://doi.org/10.1002/prot.22102): PROPKA is an empirical prediction algorithm. The disadvantage of this approach is that it works best for proteins similar to those present in the training set. Rather than checking that the later contained TWIK1-like structures, we opted for approaches based on statistical mechanics rather than a training set.

- [Citation 3 \(https://doi.org/10.1021/acs.jctc.7b00875\)](https://doi.org/10.1021/acs.jctc.7b00875): We fully agree with reviewer 1 that constant pH MD simulations, such as those developed by the Roux group and explained in the proposed citation, would be the most appropriate method to choose. In fact, one of the authors contributed recently to a paper published with B. Roux, which aims to solve the problems encountered by constant MD simulations (and other free energy-based algorithms) when the solute/solvent ratio is high. This situation is almost always observed in membrane protein simulation systems. (Galvani Offset Potential and Constant-pH Simulations of Membrane Proteins. *J Phys Chem B*, 2022. 126(36): p. 6868-6877, DOI: 10.1021/acs.jpcc.2c04593). However, for us the main drawback of cpHMD methods is their high demand on computational resources. Using cpHMD with the resources at our disposal, we would not have been able to produce enough

trajectories, which was essential to perform statistically powerful analyses. So, our choice not to use cpHMD reflects our resource management.

- Citation 1 (<https://doi.org/10.1093/nar/gki464>): We agree with reviewer 1 that H++ could have been an option satisfying the compromise outlined above. Like CHARMM, H++ calculates pKas on the basis of a Poisson-Boltzmann method or on a generalized Born method. It is therefore totally independent of a training set. In the case of a membrane protein, the H++ server requires to provide a system harbouring membrane and protein as well. This is essentially equivalent to what we did with the CHARMM algorithm. In our view, the two strategies are very similar.

The disadvantage of both approaches is that they sample a single conformation. By performing the same calculation five times, using structures extracted from short simulations in which the channel is exposed to the membrane and solvent environment, our strategy overcomes the latter problem.

4- Why is the simulated system different from that used to predict the pKa (lines 390-391)? If one repeated the same pKa predictions in this system, would their values conserved or not? How will it affect the MD simulations results?

Biological membranes have a high lipid diversity and are richer in negative species on the cytoplasmic side. Here again, our approach is based on a compromise between an oversimplified membrane (POPC only) and a “true” that is difficult to manipulate. In the same vein, we ensured that the cytoplasmic-facing leaflet was negatively charged. This additional work was carried out only the pKas had been extracted. We thank the reviewer for asking whether the updated membrane would affect pKa calculations. To enable the CHARMM PBEQ solver to calculate the pKas of a membrane inserted protein, it is possible to specify the membrane thickness and the ‘Z’-height of the protein (insertion level). To our knowledge, it is not possible to indicate lipid composition, but only an overall dielectric constant. In this context, membrane thickness is the main factor likely to affect calculations, for two reasons. Thickness influences the overall dielectric constant and, without taking into account adaptations in protein structure, a thick membrane will interact with certain residues, whereas these residues may interact with the solvent in the case of a thin membrane. We checked the thickness of the membrane. The diagram below shows that the membrane used for pKa calculations (left) and the membrane used for a few randomly selected trajectories (right) are identical. It also shows that membrane thickness neither increases nor decreases over time.

Legend: Membrane thickness used for pKa calculations and MD trajectories. The distances between the phosphorous atoms of the two leaflets are indicated. The colour codes, from light to dark green, reflect the values of the images taken at 0-10, resp. 200 ns, where the darkest corresponds to the longest simulation time.

5- The authors should comment on their results in the view of <https://doi.org/10.1016/j.bpj.2016.07.009>.

We are sorry to have forgotten to cite this important article (Oakes et al, 2016). This was done in early versions of the manuscript, and unfortunately lost during revisions between different co-authors. This paper was the first to perform MD simulations on TWIK1 and to show that residues T118 and L228, unique to TWIK1 and not found in other potassium channels, were the key to TWIK1's peculiar selectivity. A MD simulation in which H122 and D230 were protonated showed that all potassium-binding S sites in the pore were filled with water molecules, indicating that TWIK1 was the only potassium channel to exhibit such non-inactivating characteristics (inactivation of all other potassium channels leading to a non-conductive closed state). A sentence citing this work has been added in the introduction (lines 67 to 69, "*Using molecular dynamics (MD) stimulations, a previous study has highlighted the importance of Thr118 and Leu228, two residues not found in the selectivity filter of other K_{2P} channels, for the dynamic selectivity of TWIK1 (3).*"). Re-examination of this article prompted us to perform additional calculations. In this article, the authors identified highly variable distances between Asp230 and Tyr217. As Asp230 is one of the residues studied in our work, we subjected this pair of residues to our analyses. We found that the distances mentioned decreased significantly ($p < 0.01$) at low pH and added a corresponding paragraph (lines 212-218, "*In a previous MD study investigating the responses of the selectivity filter to various protonation states of His122 and Asp230, the authors observed great variability in the distance between the carboxylic acid of Asp230 and the phenolic oxygen of Tyr217, a residue located behind P2. However, their data could not identify a relationship between a protonation state and this distance [3]. According to our simulations, this distance reduces at low pH (Fig. S5), a trend also observed in the case of the H122N mutant. As Asp230 is the only residue titrated in this region, we deduce that this pH response is related to the protonation of Asp230, probably in connection with the enlargement of the S0-S1 region.) and reported this analysis in the supplementary material (Fig. S5)"). We also discussed this article in the discussion (lines 304 to 305, "*Previous MD simulations suggest that protonation of His122 and Asp230 produces TWIK1 channels in an open conformation with S sites of the selectivity filter filled with water molecules. This state of the pore would allow Na^+ transport (3).*").*

6- Conclusions are missing, which would help the reader putting together the results which might seem otherwise scattered

We added a last paragraph of conclusions, one concerning the evolution of the TWIK1 pore leading to Na^+/K^+ cotransport and the second concerning the interest of the approach based on MD simulation to study dynamic conformational changes in ion channels. This paragraph has been added in the revised manuscript (lines 330 to 335, "*In conclusion, the identification of titratable residues, electrophysiology and MD simulations have identified dynamic changes controlling the conformation of the TWIK1 pore that could not be identified by electron microscopy alone. These changes involve a network of residues in the immediate vicinity of the selectivity filter, showing that Na^+ transport may result from an evolution of the classical K^+ -selective pore. In addition, such*

an approach based on MD simulation could be of interest for studying regulations of other ion channels based on conformational changes acting on their level of activity.”)

Minor comments:

1- The description of the simulations is quite confusing. The authors should add a table in the SI resuming all the simulations of both the sections “pKa calculations” and “MD simulations” to clarify the number of replicas and the corresponding length.

2- The authors should provide the RMSD profiles of the protein in each simulation to assess the convergence of their structures.

Here we answer to these two questions at the same time. The RMSD information was missing from the submitted work and we thank the reviewer for pointing it out. We have now updated the previous time series figure (Fig. S1), adding a panel with the RMSD profiles at the two pH values and for the two sequences studied. Both sets of data are shown in the revised Fig. S1 and cited lines 436-440.

Individual RMSD values, taken at a time interval of 2 ns, are reported in the table `rmsd_values.csv`, included in the `source_data.zip` file. The table `summary_distances_angles.csv` contains the values used to produce the various panels in Fig. 4. In the same zip file, a README file explains the abbreviations contained in the two csv files. As with the pKa calculations, the method section already mentioned the duration of the simulations: 10 ns. We omitted to indicate the number of ‘rounds’. This is done now line 396.

3- Figures should be ordered as they appear in the main text, e.g. Figure 4 appears before Figure 3. Then, their quality must be improved. For example, the labels in Figure 4 are too small to be read, there is a wrong “Figure 4” on the left/bottom and its title is too generic. The same for Figs 5 and 6 where the values on the density bar should be added. Each panel must be labeled with a letter, not the entire row as the authors did. The expressions “middle panel”, “middle” should be avoided.

We apologize for the poor quality of the uploaded figures, whereas the original figures were of publication-quality. We corrected our uploading procedure. We have slightly modified the text, and the figures are now ordered in the revised text. The title of Fig. 4 was now updated to “*MD simulations reveal low pH induced conformational changes*”. We also modified Fig. 1, 3, and 4 to remove the expressions “middle panel”. We now present all the panels from Fig. 5 and Fig. 6 in a single revised Fig. 5. The densities shown are normalized data. We now indicate values between 0 and 1 on the density bar. However, for greater clarity, we have added a more detailed description of the calculation underlying the landscape representation (lines 370-374, “*Landscape representations are essentially smoothed surfaces of distance occurrences. To construct them, the distances to be studied were matched: a given His122-Glu235 distance occurs at the same time as a given Glu235-Lys246 distance, enabling us to construct grids and 2D histograms of the occurrences. These data were then smoothed using the Scipy’s splinef2d method. The Python script that generates the landscape representation and the distance files are provided with this article.*”). The `source_data.zip` folder contains a python script that generates such landscape representations and the raw data required to produce them. This now indicated (lines 460 to 463, “A `source_data.zip` file is provided with this

paper, which contains the data necessary to reconstruct the MD RMSD values, the distance and angles values reported in Fig. 4, the distances incorporated in Fig. 5A and 5C, and a Python script that generates the landscape representations.”)

4- The computational procedure used to quantify the orientation of the residues 122 should be described in the Methods section, not in the caption of Figure 4.

This description has been moved to the method section, lines 440-444.

5- I would add one or two sentences at the end of the “Discussion” that resume the main results of the present work and highlight its impact in future research.

A paragraph has been added in the revised manuscript (lines 330-335, “*In conclusion, the identification of titratable residues, electrophysiology and MD simulations have identified dynamic changes controlling the conformation of the TWIK1 pore that could not be identified by electron microscopy alone. These changes involve a network of residues in the immediate vicinity of the selectivity filter, showing that Na⁺ transport may result from an evolution of the classical K⁺-selective pore. In addition, such an approach based on MD simulation could be of interest for studying regulations of other ion channels based on conformational changes acting on their level of activity.*”)

6- In Figure 4, I would move the analysis of the side chain’s rotation to the SI because this information is implicitly included in the distance landscape representations. Moreover, I would try to represent all the panels of Figures 5 and 6 in a single figure with a structural representation on VMD that clarify the interactions reported in each panel.

It is true that the orientation of the sidechain is one of the important factors influencing the distances from its neighbours. However, the purpose of the side-chain orientation analysis was to build a basis for placing some confidence in the simulation results. We believe that such an analysis should be part of a MD publication whenever possible. The fact that our simulations were able to reproduce a couple of structural data obtained in earlier Cryo-EM experiments gives us a basis on which to build our work. These data include the reorientation of His122 and the overall elongation of the channel. We therefore feel that this analysis belongs in the main text and in one of the main figures. Fig. 5 and Fig. 6 have been fused in a single revised Fig. 5 as suggested.

Lines 251-252: the ref is missing

Line 262: “strengthen” instead of “strengthens”

Lines 268-269: the ref is missing

Line 347: missing ref for PyMol software

Line 349: the version of VMD should be specified

Line 362: clarify the dimensions of the simulation box

Line 363: the model of the water molecules is missing

Line 402: there are missing refs

Line 408: the temperature is not specified

All these suggestions concerning typos and missing references have been followed.

Reviewer 2

We thank reviewer 2 for his/her very positive comments.

Specific comments:

- It is not clear to me what the color-coding of residues in figure 2 means. I cannot find an explanation of why some residues are labeled in purple, red, black, or green and if this means anything.

The sequence alignment previously shown in Fig. 1 has been moved to the revised Fig. 2, and the colour code associated with the residues is now well explained in the legend of revised version of Fig. 2 (*"In red, the residues unique to TWIK1 and involved in ionic selectivity change. In green, the primary pH sensor in TWIK1 and TASK3. In blue, the titratable residues highlighted in this study. In bold black, residues swapped between TWIK1 and TASK3."*).

- The network of residues in figure 4 B are hard to see. Coloring the helices lighter/ making them more transparent/ increasing the size of the specific residues shown would help.

The network of residues in Fig. 4 was difficult to see due to the poor quality of the figures resulting from the pdf conversion of the manuscript. By lightening the helices and increasing the size of the highlighted residues, as well as presenting the figures in publication quality, they are now much easier to read.

Reviewer #1 (Remarks to the Author):

The revised version of the manuscript "Mechanistic basis of the dynamic response of TWIK1 ionic selectivity to pH" by Chatelain et al. has been improved as compared to the original version. However, I still have several comments that should be addressed before publication.

1. I appreciate the discussion about the methods to predict the pKa but some comparison with alternative approaches would be beneficial both to understand the advantage of "designing a new strategy" and to validate it. For example, are the pKas predicted with H++ similar to those predicted by the authors?
2. It is not easy to understand how the authors computed the RMSD profiles. What are the reference conformations? Why are there only 300 ns of simulations while the authors performed 6.3 μ s at pH 7.4 and 5.6 μ s at pH 6.0 for the WT and 2.4 μ s for the H122N mutant? Are the profiles computed using a trajectory resulting from the concatenation of all the replicas or did they refer only to a single replica? These comments refer also to the remaining panels of Figure S1. In the main text, the number of replicas and the total time of the simulations per system have been correctly specified but it is not clear which simulations of which lengths they used to do the post-processing analyses. For this reason it is important to add a table in the SI summarising all the simulations they did. It would also help to understand the data reported in the zip file.

Minor comments:

1. In order to improve the readability of the paper, the authors should explicit the logical connections between the statements. For example, at line 82 they should add their own sentence: "this figure shows that mimicking the protonation of H122 does not affect dynamic ionic selectivity, implying that other TWIK1 residues are required for the acidification-induced change in ionic selectivity". In this way, the rest of the paragraph would be clearer.
2. I suggest the authors clarify the point of lines 415-416 by adding the figure of the membrane thickness in the SI and introducing a short sentence for its discussion in the main text.
3. I appreciate that the authors put the figures in order of appearance. The same should be done with panels. For example, Figure 1C should not be before Figure 1B.
4. In the caption of Figure 2 (line 548): "titratable" instead of "titrable". Then, "important residues" for what? Why did they not show L228 that is an important residue for the ionic selectivity of TWIK1 channels?
5. In the caption of Figure 3 the first sentence should not contain references to panels.
6. In the caption of Figure 4 (line 574): "conformational" instead of "confirmational".
7. Line 373: missing ref for the Scipy's splinef2d method.

We thank the reviewer for the positive appreciation of the revised manuscript and of the changes we made. Our responses to her/his comments and suggestions are detailed below.

1. I appreciate the discussion about the methods to predict the pKa but some comparison with alternative approaches would be beneficial both to understand the advantage of "designing a new strategy" and to validate it. For example, are the pKas predicted with H++ similar to those predicted by the authors?

The structures used to calculate pKas with CHARMM have been now submitted to the H++ automated server, allowing comparison of accuracy on the same coordinates. We report the comparison in a new section in the supplementary material, which is referred in the method section (lines 395-397, *A comparison with the fast pKa automated system H++ supports the idea that the PBEQ module, although time-consuming, is sufficiently accurate for this type of project, as shown in supplementary section SI-1*). In short, we show that the CHARMM PBEQ appears to be more accurate in identifying Lys residues and confirming that His122 is a proton sensor.

2. It is not easy to understand how the authors computed the RMSD profiles. What are the reference conformations? Why are there only 300 ns of simulations while the authors performed 6.3 μ s at pH 7.4 and 5.6 μ s at pH 6.0 for the WT and 2.4 μ s for the H122N mutant? Are the profiles computed using a trajectory resulting from the concatenation of all the replicas or did they refer only to a single replica? These comments refer also to the remaining panels of Figure S1. In the main text, the number of replicas and the total time of the simulations per system have been correctly specified but it is not clear which simulations of which lengths they used to do the post-processing analyses. For this reason it is important to add a table in the SI summarising all the simulations they did. It would also help to understand the data reported in the zip file.

We thank the reviewer for encouraging us to clarify this point. Replicas were not concatenated, but values from independent simulations were averaged. The reference conformation of each trajectory was extracted after minimization and equilibration but before the plain simulation. For RMSD calculations, 14 independent trajectories were used. Every 2 ns, the conformation was compared with the reference conformation. This referencing/concatenation seems to be the soundest approach for MD time series. The lines in Fig. S1 represent the mean of the 14 calculations, and the shaded areas illustrate the standard deviations. We have revised the legend of Fig. S1 to better explain this point. (*RMSD resp. the distances were extracted every 2 resp. 10 ns and compared to the conformation obtained after minimisation and equilibration. At each pH, the RMSD values of 14 resp. 8 individual WT resp. H122N trajectories were extracted, enabling to calculate the mean time evolution of the RMSD and the error (standard deviation).*)

The lengths of these individual simulations are now shown in Table S1 of the supplementary material. The table shows that for angles and distances, 18 WT trajectories were used for each pH value, while only 14 were retained for RMSDs. Four simulations at pH 6 lasted only 200 ns long. As we wanted to show the complete evolution of RMSD for at least 300 ns, these trajectories were not used in this case. For correct balancing, four WT simulations at pH 7.4 were randomly removed. We believe that the presence of these short (200 ns) trajectories does not significantly affect the distance time series shown in Fig. S1. The new total lengths (5.8 μ s at pH 7.4 and 4.8 μ s at pH6.0) were corrected lines 429-430.

Minor comments:

1. In order to improve the readability of the paper, the authors should explicit the logical connections between the statements. For example, at line 82 they should add their own sentence: "this figure shows that mimicking the protonation of H122 does not affect dynamic ionic selectivity,

implying that other TWIK1 residues are required for the acidification-induced change in ionic selectivity". In this way, the rest of the paragraph would be clearer.

The revised version of the manuscript has been modified according to this suggestion (Lines 82-84, *This result shows that other residues are required for the acidification-induced change in ionic selectivity. To identify these residues, we first focused on the area of the selectivity filter (Fig. 2)*).

2. I suggest the authors clarify the point of lines 415-416 by adding the figure of the membrane thickness in the SI and introducing a short sentence for its discussion in the main text.

Membrane thickness is discussed and illustrated in a new section of the supplemental material. The revised manuscript has been modified accordingly (lines 424-426, *We show in SI-2 section that this membrane modification is unlikely to significantly affect the pKas of the membrane inserted protein.*).

3. I appreciate that the authors put the figures in order of appearance. The same should be done with panels. For example, Figure 1C should not be before Figure 1B.

The suggested changes have been done in revised Fig. 1 and Fig. 3, and the text of the revised manuscript modified according to these changes.

4. In the caption of Figure 2 (line 548): "titratable" instead of "titrable". Then, "important residues" for what? Why did they not show L228 that is an important residue for the ionic selectivity of TWIK1 channels?

The title of Fig. 2 has been corrected. Leucine 228 is now shown in the revised version of the Fig. 2.

5. In the caption of Figure 3 the first sentence should not contain references to panels.

References to the panels have been removed from the Fig. 3 legend.

6. In the caption of Figure 4 (line 574): "conformational" instead of "confirmational".

The typo has now been corrected.

7. Line 373: missing ref for the Scipy's splinef2d method.

A specific reference for the splinef2d was not found, probably because this method is integrated in the Scipy.interpolate sub-package. For this reason, the reference of the Python program and that of the Scipy version were added lines 374-375 (*For this, Python 3.8.5 was used, in which the sub-package scipy.interpolate from the scipy version 1.5.2 was imported*).

Reviewer #1 (Remarks to the Author):

The authors have addressed all issues raised. I thus recommend this manuscript for publication in Nature Communications.